

# Comparison of heparinized saline and 0.9% sodium chloride for maintaining central venous catheter patency in healthy dogs

Julieann Vose[1], Adesola Odunayo[1], Joshua M. Price[2], Maggie Daves[1], Julie C. Schildt[1] and M. Katherine Tolbert[3]

[1] Department of Small Animal Clinical Sciences, University of Tennessee—Knoxville, Knoxville, TN, USA
[2] Office of Information Technology, University of Tennessee—Knoxville, Knoxville, TN, USA
[3] Department of Small Animal Clinical Sciences, Texas A&M University, College Station, TX, USA

## ABSTRACT

**Background:** The purpose of this study was to determine whether heparinized saline (HS) would be more effective in maintaining the patency of central venous catheters (CVCs) in dogs compared to 0.9% sodium chloride. This was a prospective randomized blinded study conducted at a University Veterinary Teaching Hospital.

**Methods:** A total of 24 healthy purpose-bred dogs were randomized into two groups: a treatment and a control group. A CVC was placed in the jugular vein of each dog. Each dog in the treatment group had their CVC flushed with 10 IU/mL HS, while dogs in the control group had their CVC flushed with 0.9% sodium chloride every 6 h for 72 h. Immediately prior to flushing, each catheter was evaluated for patency by aspiration of blood. The catheter site was also evaluated for phlebitis, and a rectal temperature was obtained in each dog every 6 h. Prothrombin (PT) and activated partial thromboplastin (aPTT) times were evaluated prior to the administration of any flush solution. Results were then compared to values obtained 72 h later.

**Results:** All CVCs in both groups were patent after 72 h, which was demonstrated by aspiration of blood and ease of flushing the catheter. Two CVCs in the 0.9% sodium chloride group had a negative aspiration at hour 12 and 36, respectively. One CVC in the HS group had a negative aspiration at hour 18. Signs of phlebitis occurred in three dog: two in the 0.9% sodium chloride group and one in the HS group. No dog was hyperthermic (>103 °F). Two catheters were inadvertently removed by dogs in the HS group during the study. There were no significant differences in catheter patency, incidence of phlebitis, or incidence of negative aspirations between both groups. aPTT and PT values remained within the normal reference range for all dogs in both groups. Ultimately, 0.9% sodium chloride was as effective as 10 IU/mL HS in maintaining the patency of CVCs for up to 72 h in healthy dogs. Further evaluation in clinical patients is warranted.

Corresponding author
Adesola Odunayo,
aodunayo@utk.edu

## INTRODUCTION

Central venous catheterization was first performed in 1929 and has become a mainstay of modern clinical practice (*Smith & Nolan, 2013*).They are indispensable devices utilized in both human and veterinary medicine to deliver intermittent medications, continuous intravenous infusions, obtain serial blood draws, provide parental nutrition, aid in hemodynamic monitoring and interventions, and provide access for extracorporeal blood circuits (*Elliott, Fleeman & Rand, 2010*; *Smith & Nolan, 2013*; *Tan, Dart & Dowling, 2003*; *Ueda, Odunayo & Mann, 2013*).

Complications associated with indwelling central venous catheters (CVCs) include phlebitis, occlusion, infection at the catheter site, catheter related blood stream infection, and thrombosis (*Parienti et al., 2015*). Replacement of CVCs often leads to increased patient discomfort, increased need of sedation, and increased client cost (*Cowl et al., 2000*; *Goode et al., 1991*; *LeDuc, 1997*). Although heparin is an effective anticoagulant, its use in preventing thrombosis and maintaining CVC patency has been controversial in human patients (*Zhong et al., 2017*). While a few human studies showed heparin as more effective at maintaining CVC patency than 0.9% sodium chloride, in many others a demonstrable advantage of heparin over 0.9% sodium chloride was not observed (*Bradford, Edwards & Chan, 2015*; *Klein et al., 2018*; *López-Briz et al., 2014*; *Schallom et al., 2012*; *Zhong et al., 2017*). A recent Cochrane review did not find a demonstrable benefit of heparin over 0.9% sodium chloride, although the quality of evidence studied was low (*López-Briz et al., 2018*).

Complications associated with the use of heparinized saline (HS) include a potentially fatal complication called Heparin-induced Thrombocytopenia (HIT) (*McNulty, Katz & Kim, 2005*). HIT is an immunological response to heparin in which platelets are activated, causing platelet clumping, and clearance from the body (*Ahmed, Majeed & Powell, 2007*; *Mian et al., 2017*). To the authors' knowledge, HIT has not been documented in dogs. Other complications of heparin use include hypocoagulability, heparin-induced hyperkalemia; allergic reactions; and drug interactions (*Mitchell et al., 2009*). HS also tends to be more expensive to procure thus increasing the cost of medical care. The process of making HS in veterinary hospitals may also potentially increase the risks of flush contamination. Thus, many human hospitals have discontinued the use of heparinized flushes in hospitalized patients based on consensus statements or guidelines (*Ahmed, Majeed & Powell, 2007*; *Barrett & Lester, 1990*; *Frykholm et al., 2014*). To the authors' current knowledge, there are no studies evaluating the use of HS and 0.9% sodium chloride in maintaining patency in CVCs in dogs.

The purpose of this study was to compare two flush solutions in maintaining patency in CVCs placed in dogs compared to 0.9% sodium chloride. A second objective was to determine if the use of HS caused changes in the aPTT values of healthy dogs. The hypothesis of this study is that 0.9% sodium chloride would be equally effective at maintaining CVCs as 10 IU/mL HS in healthy dogs.

## MATERIALS AND METHODS

The present study was a prospective, randomized, double-blinded study approved by the Institutional Animal Use and Care Committee at the University of Tennessee

(Protocol #2518). Healthy purpose-bred research dogs were utilized in this study. All dogs were deemed healthy based on their physical examination findings, as well as results of a complete blood count, serum biochemical profile, and urinalysis. The dogs were then randomized into two group: the HS or 0.9% Sodium Chloride (S) flush group.

Butorphanol (0.4 mg/kg), midazolam (0.4 mg/kg), ketamine (two mg/kg), and dexmedetomidine (four µg/kg) were administered intravenously through a cephalic vein for sedation. Oxygen supplementation was then provided and physiologic parameters (heart rate, temperature, respiratory rate, electrocardiogram, and blood pressure) were monitored. The dogs were placed in left lateral recumbency, and the area around the right jugular vein was shaved and aseptically prepared. A 14 Ga 20 cm polyurethane single-lumen CVC[a], with a priming volume of 0.35 mL, was placed using the Seldinger technique as described elsewhere (*Aspinall & Aspinall, 2013*). When the CVC could not be successfully placed in the right jugular vein, the left jugular vein was utilized. Two view thoracic radiographs were taken to confirm ideal catheter placement.

Blood was obtained from the CVC immediately after placement and before flushing for assessment of the prothrombin time (PT) and activated partial thromboplastin time (aPTT). The CVC was secured with sutures, flushed with either HS or S, and a non-adherent dressing[b] was placed at the insertion site with a bandage applied over the dressing. Needle-free connectors[c] were utilized for all catheters. The dogs were then fitted with an Elizabethan collar after recovery from sedation.

### Catheter evaluation

The CVCs were evaluated by one of three investigators (JV, MD, or AO). Every dog had its catheter site evaluated for evidence of phlebitis every 6 h by shifting the neck bandage down and exposing the CVC insertion site. Phlebitis was defined as the presence of any of the following: erythema, tenderness, swelling, unusual discharge, or warmth. A rectal temperature was also obtained from each dog during each evaluation. Hyperthermia was defined as a rectal temperature greater than 103 °F.

Catheter patency was evaluated as the ability to aspirate at least 0.5 mL of blood from the catheter with little resistance. Catheter patency was also assessed by the ability to administer three mL of flush solution with little resistance through the CVC. The flush solution was administered only after the CVC had been evaluated for phlebitis and aspiration of blood had been attempted.

All CVCs in the HS and S group were evaluated for phlebitis and patency in the same manner.

### Catheter flushes

All CVC flushes were performed by one of three investigators (JV, MD, or AO). The study solutions were prepared by a veterinary technician who was not a part of the study. Each syringe of the study solution was labeled with a predetermined study code to ensure blinded randomization. The dogs in the HS and S groups had their CVCs evaluated and flushed with three mL of 10 IU/mL HS and three mL of 0.9% sodium chloride, respectively, every 6 h for 72 h. The catheters were flushed using one quick push of the

[a] Arrow central venous catheters. Teleflex, Morrisville, NC, USA

[b] Telfa. Cardinal Health, Dublin, OH, USA

[c] MILA international, Inc, Florence, KY, USA

flush solution, over about 3–5 s and the catheters were immediately clamped after the flush was completed. Once enrolled, the subjects continued to receive the same study solution throughout the duration of the study. Catheters were removed upon discovery of non-patency, inadvertent removal by the dog, or at study completion.

### Evaluation of hypocoagulability and catheter care

The CVC insertion site was cleaned with chlorohexidine scrub every 24 h. The non-adherent dressing and neck wrap were also replaced at this time. The dogs were observed by one of the investigators (JV, MD, or AO) every 3 h to ensure the CVCs had not been inadvertently removed. Just before the CVC was removed at the end of the study, blood was obtained to evaluate the PT, and aPTT for each dog.

### Statistical analysis

A power analysis calculation was performed prior to the study to determine an appropriate sample size. It was determined that 24 dogs would be a sufficient population to show a difference in catheter occlusion between the HS and S group with a power of 0.80 of an alpha <0.05. Data was analyzed using commercial statistical software.[d] A Fisher's exact test was used to analyze whether or not there was a significant overall relationship between whether each dog had an occurrence of phlebitis and the treatment solution they received. A generalized linear mixed model with a binary distribution and logit link function was used to evaluate the occurrence of both phlebitis between the two treatment groups over time. A repeated measure mixed effects ANOVA was used to evaluate the differences in temperature, PT, and PTT values between treatment solutions over time and the solution by time interaction; Holm's p-value adjustment method was applied to post hoc tests. Shapiro–Wilk normality tests and QQ plots were used to evaluate normality of mixed model ANOVA residuals. Levene's equality of variances test was used to evaluate equality of variances between treatments. All statistical assumptions regarding normality and equality of variances were met.

## RESULTS

A total of 24 purpose-bred dogs were enrolled in the study. A total of 12 dogs were randomly assigned to the S group, consisting of four intact males and eight intact females while 12 dogs were assigned to the HS group, consisting of five intact males, and seven intact females. The mean age of the dogs was 20.33 months (10–69 months). The average weight of the S and HS groups was 21.25 kg (SD ± 5.41 kg) and 13.92 ± 4.07 kg, respectively. A total of 10 catheters were placed in the right jugular vein, while two catheters were placed in the left jugular vein. Both catheters placed in the left jugular vein were accidentally placed in the left jugular due to a communication error during placement. All CVCs were placed during the first attempt without any complications associated with the technical procedure. Two dogs experienced single occasional ventricular premature contractions during catheter placement that resolved without intervention. There were no other complications noted during catheter placement. All catheters terminated at the cranial vena cava on the post placement radiographs.

[d] SAS Institute Inc., SAS Campus Drive, Cary, North Carolina 27513, USA

**Table 1 Mean rectal temperature over time in 24 healthy dogs with 0.9% sodium chloride or heparinized saline flushes.**

| Hour | 0.9% Sodium chloride (S) | 10 IU/mL Heparinized saline (HS) |
|---|---|---|
| 0 | $99.00 \pm 1.52^{D}$ | $99.00 \pm 1.41^{D}$ |
| 6 | $100.39 \pm 1.08^{ABC}$ | $100.58 \pm 0.51^{ABC}$ |
| 12 | $100.57 \pm 0.91^{BC}$ | $100.23 \pm 0.48^{BC}$ |
| 18 | $100.62 \pm 0.69^{ABC}$ | $100.53 \pm 0.94^{ABC}$ |
| 24 | $100.74 \pm 0.68^{A}$ | $101.07 \pm 0.77^{A}$ |
| 30 | $100.60 \pm 1.05^{ABC}$ | $100.36 \pm 0.50^{ABC}$ |
| 36 | $100.45 \pm 0.80^{BC}$ | $100.10 \pm 0.44^{BC}$ |
| 42 | $100.3 \pm 0.49^{BC}$ | $100.39 \pm 0.86^{BC}$ |
| 48 | $100.58 \pm 0.47^{AB}$ | $100.73 \pm 0.73^{AB}$ |
| 54 | $100.15 \pm 0.34^{C}$ | $100.13 \pm 0.50^{C}$ |
| 60 | $100.25 \pm 0.61^{C}$ | $99.90 \pm 0.34^{C}$ |
| 66 | $100.20 \pm 0.91^{CD}$ | $99.77 \pm 0.35^{CD}$ |
| 72 | $100.77 \pm 0.74^{AB}$ | $100.76 \pm 0.57^{AB}$ |

Note:
Mean rectal temperature and standard deviation of 24 healthy dogs with 0.9% sodium chloride ($n = 12$) or heparinized saline ($n = 12$) central venous catheter flushes. Significant differences were observed over time ($p < 0.0001$) but not between treatments. Mean rectal temperatures that do not share a common superscript letter differed significantly ($p < 0.05$) based on Holm's post hoc analysis.

All CVCs in both groups had catheters that flushed easily at every time point of evaluation. Two of the CVCs in the S group did not aspirate blood back, one at 12 h and the other at 36 h, both of which were positive for aspiration of blood at subsequent evaluations. One CVC in the HS group did not aspirate blood back at 18 h but was positive for aspiration at subsequent evaluations. There was no significant difference in the odds of aspirating blood between the HS and S treatment groups or over time ($p = 0.70$ and $p = 0.97$, respectively). In the HS group, the study was discontinued after 48 h in one dog and 54 h in another dog, due to inadvertent catheter removal by both dogs.

Subjective signs of mild phlebitis were observed in two dogs in the S group at 54 and 72 h, respectively, and one from the HS group at 48 h. There was no evidence of ongoing inflammation or patient discomfort, so those CVCs were maintained until the end of the study. However, there was no significant difference in the odds of phlebitis occurring between groups or over time ($p = 0.65$ and $p = 0.68$, respectively).

There was no evidence of hyperthermia recorded in any dog in either the S or HS group. No significant difference in mean temperature between groups was observed ($p = 0.68$) (Table 1). However, there was a difference in mean temperature over time in both groups ($p < 0.0001$). The post hoc test revealed a significant increase in mean temperature of approximately 1.5° between initial observations at time 0 and 6 h into observation ($p = 0.0009$). A significant notable mean increase of approximately 1.8° was observed between initial observations at time 0 and the conclusion of the study after 72 h ($p < 0.0001$). The maximum mean difference over time of approximately 1.9° occurred between initial observations at time 0 and 24 h into observation ($p < 0.0001$).

**Table 2  Mean baseline (T0) and 72 h (T72) PT and aPTT values of 24 healthy dogs who either had their central venous catheters flushed with 0.9% sodium chloride group (*n* = 12) or heparinized saline (*n* = 12).**

|  | Mean PT, (s) | Mean aPTT, (s) |
|---|---|---|
| **0.9% Sodium chloride T0** | 8.45 ± 0.57[*],[‡] | 15.49 ± 2.23[†],[§] |
| **0.9% Sodium chloride T72** | 8.26 ± 0.41[*] | 17.06 ± 3.87[†] |
| **10 U/mL Heparinized saline T0** | 8.64 ± 1.79[*],[‡] | 18.50 ± 5.66[†],[§] |
| **10 U/mL Heparinized saline T72** | 8.06 ± 0.51[*] | 19.12 ± 2.57[†] |

Notes:
[*] $p = 0.03$.
[†] $p = 0.003$.
[‡] $p = 0.676$.
[§] $p = 0.069$.
PT, prothrombin time; aPTT, activated partial thromboplastin time.

The PT and aPTT remained within reference ranges for all dogs at all time points (Table 2). There was no significant difference between PT or aPTT values in either treatment group ($p = 0.67$ and $p = 0.07$, respectively). Significant pre-test, post-test differences were observed for both PT and aPTT regardless of treatment received ($p = 0.03$ and $p = 0.003$, respectively). Mean PT values decreased by approximately 0.15 s, while aPTT values increased by approximately 1.9 s between pre and post measures. There was no evidence of spontaneous or unexplained bleeding in any dog in either group. No CVC had to be removed prematurely because of non-patency, and all CVCs were successfully removed at the end of the study.

## DISCUSSION

Despite the overwhelming evidence in human patients showing that 0.9% sodium chloride is an effective flushing solution to maintain CVC patency (*Bradford, Edwards & Chan, 2015, 2016; Heidari Gorji et al., 2015; LeDuc, 1997; Ziyaeifard et al., 2015*), HS is still used in many veterinary facilities (*Fleeman, 2001*). Based on the results of the present study, 0.9% sodium chloride appears to be a safe alternative to heparin for flushing CVCs and may prove to be more cost effective for facilities, saving both time and money (*Ridyard et al., 2017*). A previous study evaluating the ideal flush solution for peripheral catheters in dogs had similar findings as the present study (*Ueda, Odunayo & Mann, 2013*). A potential reason there was no observable difference between flush solutions used in the present study might be due to the fact that the physical act of flushing (the technique used and the flush volume utilized) plays the most important role in maintaining catheter patency, while the chemical properties of the flush solution itself might play a negligible role (*Goossens, 2015*). Flushing with a turbulent technique (push-pause or pulsatile) enhances the rinsing effect in the catheter and helps prevent catheter malfunction (*Goossens, 2015*). The present study used the quick push technique and its efficacy in preventing catheter malfunction is currently unknown.

Factors associated with the development of intraluminal thrombosis in CVCs include increased length of catheter duration, catheter size, catheter-related sepsis, and infusion of blood products (*Park et al., 2014; Timsit et al., 1998*). In human patients, symptomatic catheter-related central vein thrombosis is infrequent but asymptomatic catheter-related

central vein thrombosis (diagnosed by venographic studies) may be as high as 33% (*Timsit et al., 1998*). The use of prophylactic anticoagulants, like heparin, to prevent catheter thrombosis has been studied extensively in critically ill human patients (*Akl et al., 2011*; *Dal Molin et al., 2014*; *Kirkpatrick et al., 2007*; *Schallom et al., 2012*; *Vidal et al., 2014*). In some studies, systemic anticoagulation of the patient may help reduce thromboembolic disease. However, there is no strong evidence in human patients that heparin flushes reduce the incidence of intraluminal thrombosis in CVCs. Indeed, anticoagulant prophylaxis to prevent intraluminal CVC thrombosis is not recommended in human patients (*Farge et al., 2013*; *Schallom et al., 2012*; *Smith & Nolan, 2013*).

Three catheters in the present study had one episode of inability to withdraw blood during one evaluation point, although all three catheters flushed easily at each of those time points. It is likely that the inability to withdraw blood had to do with the position of the dog's neck or mechanical kinking, which may alter the position of the tip of the CVC and potentially interfere with the ability to withdraw blood (*Stephens, Haire & Kotulak, 1995*). However, it is also possible that all three catheters had intraluminal thrombosis, which inhibited blood withdrawal, also known as the "ball valve" effect. The "ball valve" effect is thought to be caused by fibrin obstructing the catheter tip when negative pressure is applied in an attempt to withdraw blood but does not obstruct the flow of flush solution through it (*Stephens, Haire & Kotulak, 1995*). However, in human patients, the inability to withdraw blood does not predict the presence of intraluminal thrombosis (*Kuter, 2004*). In one study, 57% of thrombosed CVCs compared to 27% of non-thrombosed CVCs failed to allow blood to be drawn (*Gould, Carloss & Skinner, 1993*; *Kuter, 2004*). In a venographic study of CVCs that had problems with blood withdrawal in human patients, 58% had thrombosis, and 42% did not have thrombosis. They concluded that non-thrombotic mechanical problems commonly prevent blood flow (*Kuter, 2004*; *Stephens, Haire & Kotulak, 1995*). Advanced imaging was not used to determine the reason blood was unable to be withdrawn from the three catheters in this study but may be considered in future studies.

Apart from thrombosis, other CVC related complications reported in human patients include catheter occlusion, CVC blood stream infections, air embolism, catheter obstruction, infection of the insertion site, pulmonary thromboembolism, deep vein thrombosis, and mechanical complications (*Blaiset et al., 1995*; *Parienti et al., 2015*). The present study was designed to evaluate for local inflammation (phlebitis and evidence of infection at the insertion site) and, to a degree, systemic infection (fever, lethargy, vomiting, anorexia, or change in attitude of the dogs). There was evidence of local inflammation (phlebitis) in this study, all of which happened after 48 h of catheterization with no protective value of heparinized flushes on local inflammation. There was no evidence of systemic infection in this study, although this might be due to the short duration of the study. In a recent study evaluating complications associated with CVCs in 47 dogs and cats, only one patient developed signs of inflammation (phlebitis) (*Reminga et al., 2018*). This patient also had an unexplained fever in conjunction with the local signs of inflammation (thickening of the vessel, erythema, and discomfort). In the present study, there were no localized catheter-related infection identified. It is possible that

phlebitis is an uncommon complication of CVCs in veterinary patients, although further studies are warranted to better evaluate this.

There was no evidence of hyperthermia in the present study. The study plan was to evaluate the body temperature of the dogs to see if there might be an increased temperature correlating to the development of phlebitis. There was a very low incidence of phlebitis in the present study, and no dog had any evidence of systemic infection. The difference in temperature over time is largely attributed to mild decreases in temperature following sedation and rebound normothermia once the dogs were awake.

Another potential complication of heparinized flush is the development of hypocoagubility and spontaneous hemorrhage after its continued use. In one case report, a human patient experienced postoperative bleeding as a result from overuse of "heparin flushes" and required a second surgery (*Passannante & Macik, 1988*). Another objective of the study was to set out to determine if dogs were at risk for spontaneous bleeding due to the use of heparinized flush. There were no statistically significant differences in the change in aPTT (which is primarily affected with the use of heparin) and PT when the HS and S groups were compared. The significant differences between the PT and aPTT over time, regardless of treatment received, is difficult to explain. These differences are not thought to be clinically significant. However, the mean weight of dogs in this study was 17.58 kg, and the catheters were only flushed four times a day. Critically ill veterinary patients often have their CVCs flushed frequently during the day due to administration of medications or blood draws from the catheter. Thus, the effect of heparinized flushes on the coagulation times of smaller sized dogs, as well as after more frequent flushes, still needs to be evaluated.

In the present study, two dogs experienced ventricular premature contractions during CVC placement. This was likely because the guide wire was in close association with the right atrium during catheter placement. Both dogs recovered uneventfully, and their arrhythmia resolved without therapeutic intervention, once the guidewire was backed out of the dog. Ventricular arrhythmias are commonly reported arrhythmias during CVC placement in human and veterinary patients and generally tend to be self-limiting, although fatal arrhythmias have been reported in human patients (*Nayeemuddin, Pherwani & Asquith, 2013*; *Reminga et al., 2018*; *Vesely & Radiology, 2003*; *Yilmazlar et al., 1997*). Other potential causes of ventricular premature contractions in these dogs may include hypoxemia, underlying heart disease or as a result of the sedative administered.

One important limitation of this study is that post hoc observed power calculations resulted in the determination that this study somewhat lacked power to detect differences between treatments. This is likely because there was limited incidence of catheter dysfunction in both groups. While the study was adequately powered to determine differences over time, there was limited power to detect differences between treatment groups. Another limitation of the study is the fact that the study was done in healthy dogs. Critically ill dogs may have underlying alterations in their coagulation system, making them more susceptible to thrombotic events in their CVCs. However, in many studies performed in critically ill human patients, a protective effect of heparin against catheter thrombosis has not been identified. Additional studies, evaluating

the protective effects of HS, should be performed in critically ill dogs. An additional limitation of the current study is the observation period. A duration of 72 h was chosen based on the subjective duration of CVCs in our institution. *Reminga et al. (2018)* also reported a median duration of 3 days for critically ill dogs and cats with CVCs in their institution. It is possible that a protective effect of heparin will be seen if the study was performed over a longer period since *Reminga et al. (2018)* concluded that longer catheter dwell times was a risk factor for indwelling complications. Mechanical obstructions (venous thrombosis, kinking) were the most common cause of CVC failure in *Reminga et al.'s (2018)* study (7/47, 14%); however, they did not specify how many catheters had thrombosis vs. kinking, neither did they specify the flush solution used at their institution. The final limitation of the present study is that the investigators who flushed the catheters in the present study subjectively determined the patency of the CVCs. While this is not an objective way of determining catheter patency, it is the same method utilized in clinical patients, both in human and veterinary medicine. Moreover, the investigators were blinded to treatment.

## CONCLUSIONS

In conclusion, 0.9% sodium chloride flushes were found to be as effective as 10 IU/mL HS flushes in maintaining the patency of CVCs in healthy dogs for up to 72 h.

## ACKNOWLEDGEMENTS

The authors would like to thank Amanda Hand, MA; Tamberlyn Moyer, LVMT, VTS (Nutrition); Jimmy Hayes, LVMT; Skyler Bowers, LVMT; and Rachel Feuerstein, LVMT for their expertise and technical support.

### Funding

This project was funded by the University of Tennessee's Center of Excellence Fund. The funders had no role in study design, data collection and analysis, decision to publish, or preparation of the manuscript.

### Grant Disclosures

The following grant information was disclosed by the authors:
University of Tennessee's Center of Excellence Fund.

### Competing Interests

The authors declare that they have no competing interests.

### Author Contributions

- Julieann Vose conceived and designed the experiments, performed the experiments, analyzed the data, contributed reagents/materials/analysis tools, prepared figures and/or tables, authored or reviewed drafts of the paper, approved the final draft.

- Adesola Odunayo conceived and designed the experiments, performed the experiments, analyzed the data, contributed reagents/materials/analysis tools, prepared figures and/or tables, authored or reviewed drafts of the paper, approved the final draft.
- Joshua M. Price conceived and designed the experiments, analyzed the data, contributed reagents/materials/analysis tools, prepared figures and/or tables, authored or reviewed drafts of the paper, approved the final draft.
- Maggie Daves performed the experiments, analyzed the data, contributed reagents/materials/analysis tools, approved the final draft.
- Julie C. Schildt performed the experiments, contributed reagents/materials/analysis tools, approved the final draft.
- M. Katherine Tolbert analyzed the data, contributed reagents/materials/analysis tools, approved the final draft.

## Animal Ethics

The following information was supplied relating to ethical approvals (i.e., approving body and any reference numbers):

University of Tennessee Institutional Animal Care and Use Committee provided full approval for this research (#2518).

## Data Availability

The raw data for the study is available in the Supplemental File. The raw data sheet describes the case number of each dog, sex, type of solution used, temperature recorded every 6 h, as well as if blood could be aspirated from the catheter, if the catheter was patent, and if there was evidence of phlebitis at the catheter site.

## Supplemental Information

Supplemental information for this article can be found online at http://dx.doi.org/10.7717/peerj.7072#supplemental-information.

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
