# Peer review of "Comparison of heparinized saline and 0.9% sodium chloride for maintaining central venous catheter patency in healthy dogs"

_PeerJ, doi:10.7717/peerj.7072_

## Round 0.1 · original submission · Major Revisions

Both reviewers have provided detailed comments requiring careful address. In particular, please provide information on how the number of cases being assessed was derived.

·

Basic reporting

- Please consider shortening the length of the Introduction to focus on topics pertinent to this study including catheter complications and NS versus HS pros / cons.
- Please ensure that all studies are clearly indicated as human or veterinary to avoid confusion.
- Please note that one of the references is listed twice (López‐Briz E, Ruiz Garcia V, Cabello JB, Bort‐Marti S, Carbonell Sanchis R, Burls A. 2014. Heparin versus 0.9% sodium chloride intermittent flushing for prevention of occlusion in central venous catheters in adults. Cochrane Database of Systematic Reviews. 10.1002/14651858.CD008462.pub2)

Experimental design

- Please expand on the justification behind the power analysis: what was the difference that you were assessing for between the control and HS groups (i.e., lack of patency?
prolongation on clotting times? incidence of phlebitis?). Considering that human meta-analyses including >2000 human patients are insufficiently powered to determine a difference in HS versus saline for maintaining catheter patency, it seems unlikely that 24 healthy dogs would be enough to determine this.
- Please specify the type and size of CVC placed. The two incidents of ventricular arrhythmias suggest that the CVC may have been placed too far distal.
- Please explain how proper placement (position of distal end of the CVC) was determined.

Validity of the findings

- While many studies in the human literature suggest that NS and HS flushes are of equal efficacy for preserving CVC patency, recent meta-analyses suggest that the studies are largely underpowered and the conclusions remain uncertain (https://www.ncbi.nlm.nih.gov/pubmed/30058070). Please adjust your Discussion and Conclusions accordingly.

Additional comments

- Please avoid use of first-person language such as “we” or “our” throughout the manuscript.
- Please avoid statements suggesting novelty of research, since it is not possible to know what similar studies other researchers may have completed and not yet published.
- The second paragraph of the Discussion seems slightly out of place or irrelevant to this study’s objectives. Please consider revising to focus on the study objectives (i.e., maintenance of catheter patency / reduction of complications rather than to reduce systemic thromboembolic complications [not assessed by the present study]).
- In the third paragraph of the Discussion, it could be reiterated that advanced imaging was not used to determine the cause of the inability to aspirate blood from the catheter.
- Please move the discussion re: the statistical differences in temperature over time further to the 4th paragraph.
- Line 331: Would APCs be a more logical consequence of distal guidewire placement? Please suggest other possible causes of VPCs in these dogs.
- Line 343: This sentence requires re-wording for clarity since numerous limitations are mentioned.
- Please clarify that the results are applicable to healthy dogs and that a similar investigation is required in critically ill dogs before clinical recommendations can be made.
- Table 1: Please consider removing this table as it does not add anything to the manuscript that is not already stated in the text.
- Tables 2 and 3: Please specify “healthy” dogs.

Reviewer 2 ·

Basic reporting

This is a well done study and provides an answer to a long sought after question. Thanks!

1. Consider changing purpose to "comparing two flush solutions" instead of determining whether HS was "more effective".
2. Be more consistent with 0.9% sodium chloride vs. normal saline.
3. Line 122: consider deleting "as determined by prolongation of the aPTT" - unnecessary
4. Line 128-129: This sentence is too vague. The references cited simply shows similar CVC patency comparing HS and S.
5. Line 236: "regardless of the treatment each patient received" is confusing. Change to "in both groups" or "for all dogs"
6. Line 257: Can you find a more recent reference than Fleeman, 2001?
7. Line 308: "no evidence of systemic infection" is too broad of a statement since the study only took into account temperature and clinical signs.
8. I do not know the mechanism for VPCs due to the guide wire tickling the atrium although I agree it is seen. Any references you can cite?

Experimental design

1. Since reflux may contribute to CVC thrombosis formation, please specify what type of catheter plug/connector was used such as needle-free connectors.
2. As mentioned in the discussion, flushing technique may impact CVC patency. Please specify method used.
3. Very minor, but please specify if CVC lumens were clamped between aspirations/flushes.
4. couldn't find the footnote for the exact type of catheter placed.
5. Would strengthen results to give priming volume of catheter and comment on chosen volume of flush.

Validity of the findings

no comment

Additional comments

Great job! Thanks for doing this study.

---

## Round 0.2 · Minor Revisions

Both reviewers reviewers identified lack of insertion of the appropriate number in lieu of XXX, and several mis-spellings or mis-formats.

We look forward to your address of these concerns.

·

Basic reporting

No Comment

Experimental design

No Comment

Validity of the findings

No Comment

Additional comments

Dear Authors - Thank you for adequately addressing the concerns raised during review of your initial manuscript. All of my concerns have been addressed and incorporated into your revised manuscript. A remaining edit, however, is at Line 360 where it has yet to be specified (currently listed as XXX) what number of dogs would be required to assess the difference in catheter occlusion between HS versus NS in dogs. Further to that, there are no outstanding concerns. Thank you and well done!

Reviewer 2 ·

Basic reporting

Thank you for the edits. The paper is stronger as a result.

1. Line 340: Consider adding "(post-hoc)" in between "observed power" for clarity.
1. Line 344: "XXX" should be filled in
2. Line 438: All CAPS of authors should be changed.

Experimental design

Thank you for providing further clarification regarding your methods.

1. Although there is a lack of clinical evidence, guidelines usually recommend a flush volume of at least 2x the priming volume.
-Royal College of Nursing. (2010) Standards for infusion therapy. London, Royal College of Nursing.
-Infusion Nurses Society. Infusion nursing standards of practice. Journal of Infusion Nursing. 2011;34(1, supplement):S1–S110. doi: 10.1097/01.nan.0000393791.46613.51.

Validity of the findings

no comment

Additional comments

no comment

---

## Round 0.3 · accepted · Accept

Thank you for addressing the remaining concerns.